# Multi-Modality, Multi-Dimensional Characterization of Pediatric Non-Alcoholic Fatty Liver Disease

**DOI:** 10.3390/metabo13080929

**Published:** 2023-08-08

**Authors:** Neema Jamshidi, Alborz Feizi, Claude B. Sirlin, Joel E. Lavine, Michael D. Kuo

**Affiliations:** 1Department of Radiological Sciences, David Geffen School of Medicine, University of California, Los Angeles, CA 90095, USA; 2School of Medicine, Yale University, New Haven, CT 06520, USA; 3Department of Radiology, University of California, San Diego, CA 92093, USA; 4Department of Pediatrics, Columbia University, New York, NY 10027, USA; 5Medical AI Laboratory Program, The University of Hong Kong, Hong Kong SAR, China

**Keywords:** metabolomics, biological networks, systems analysis, magnetic resonance imaging, multimodal correlation network

## Abstract

Non-alcoholic fatty liver disease is a multifaceted disease that progresses through multiple phases; it involves metabolic as well as structural changes. These alterations can be measured directly or indirectly through blood, non-invasive imaging, and/or tissue analyses. While some studies have evaluated the correlations between two sets of measurements (e.g., histopathology with cross-sectional imaging or blood biomarkers), the interrelationships, if any, among histopathology, clinical blood profiles, cross-sectional imaging, and metabolomics in a pediatric cohort remain unknown. We created a multiparametric clinical MRI–histopathologic NMR network map of pediatric NAFLD through multimodal correlation networks, in order to gain insight into how these different sets of measurements are related. We found that leptin and other blood markers were correlated with many other measurements; however, upon filtering out the blood biomarkers, the network was decomposed into three independent hubs centered around histopathological features, each with associated MRI and plasma metabolites. These multi-modality maps could serve as a framework for characterizing disease status and progression and could potentially guide medical interventions.

## 1. Introduction

Non-alcoholic fatty liver disease (NAFLD) remains a significant issue, with prevalence nearing 10%; the percentage is reportedly as high as 34% in pediatric obesity clinics [1]. It is the leading cause of liver disease in children [2]. Although NAFLD is ultimately a disease of hepatic dysfunction, its development and progression are systemic, involving multiple organs [3]. Blood enzymes and proteins are the mainstays of the clinical measurements used for the management of the disease; however, the serum’s small metabolite profiling likely offers a rich yet incompletely characterized means for assessing disease severity and treatment response [4,5].

Histopathology presents the gold standard for diagnosis [6] and information gained from liver biopsies can be used to systematically characterize architectural and structural features, such as ballooning, fibrosis, and the degree of steatosis, summarized in non-alcoholic steatohepatitis (NAS) scores [7]. Even when relying on the histopathological diagnosis, the classification and subtypes of non-alcoholic steatosis continue to expand, with a notable differentiation between adult and pediatric subtypes [8]. Although liver biopsies are relatively low-risk procedures, they are not ideal for tracking the progression of disease over time or following treatment response longitudinally. Hence, there has been a need for non-invasive methods of monitoring disease status. As the diagnosis and classification of NAFLD has evolved in past decades, magnetic resonance imaging (MRI) has emerged as a powerful non-invasive means of disease characterization, particularly steatosis [9,10]. Many studies have understandably focused on lipid-related changes in NAFLD and NASH [11,12], but given the metabolic and structural changes involved in the progression of NAFLD, the metabolic alterations are unlikely to be strictly confined to a single area of metabolism. The human plasma metabolome is profoundly complex [13], but if the underlying changes in the metabolomic profiles can be unraveled, the potential for diagnosing, tracking, and treating diseases that involve metabolic derangements is profound [4]. Small molecular metabolomics can provide more nuanced changes with regard to systemic metabolism compared to individual proteins or enzymes; although the complexity and significance of the different patterns of metabolomic alterations are not yet fully understood, more studies are beginning to decipher these patterns in NAFLD/NASH [14].

Since different measurement modalities are used to diagnose and track the course of this disease and its treatment (in particular, tissue biopsies, blood enzymes, and MRI), it is non-trivial to infer how these different types of measurements are interconnected. The lack of knowledge about how different measurement modalities are related is a gap that can be filled through an integrative, system-based analysis; this highlights the need to explore the potential relationships between measurement modalities in order to better appreciate the extent to which they can detect and track the progression of the disease using a systems medicine approach [15,16]. We sought a systems view of NAFLD by integrating deep sets of disparate measurements along four different axes: clinical (blood biomarkers), multiparametric MRI, multi-dimensional histopathology, and small molecule plasma metabolomics in a pediatric NAFLD cohort (Figure 1). These datasets were integrated through the construction of multimodal correlation networks (MCNs). MCNs provide a means for integrating disparate data types; graphical exploration of these networks can provide insights into how metabolomics, structural physiology (MRI), tissue architecture (histopathology), and standard clinical measurements are interrelated in NAFLD.

## 2. Materials and Methods

This study involves a prospective analysis (i.e., plasma NMR profiling) of a retrospective cohort of pediatric subjects (clinical blood profiling, such as liver function tests, and MRIs).

### 2.1. Samples

Histopathological and clinical characterizations of liver biopsies (*n* = 65 across 18 variable measurements), liver MR imaging (*n* = 26 across 6 variable measurements), and NMR plasma profiling (*n* = 48 across 30 metabolite measurements) of children (8 to 17 years old), were conducted with parental consent and subject assent (in accordance with the Declarations of Helsinki and Istanbul; the review and approval of the protocol were under UCSD IRB number 050377). None of the participants took experimental medications or were enrolled in any other study at the time of the sample acquisition. Venipuncture fasting blood samples were obtained, spun down for 10 min at 3000 rotations per minute at 20 °C, and further processed as previously described [17].

### 2.2. Experimental Measurements

All subjects were confirmed to have NAFLD via biopsy and underwent liver MR imaging, which involved the calculation of hepatic fat fraction and the use of Liver Imaging of Phase-Interference Signal Oscillation and Quantification (LIPO-Quant), derived from a multi-peak fat spectral model averaging values from three separate regions of interest, as previously described [18], in addition to diffusion-weighted spin-echo echo-planar and T2 single-shot fast spin-echo images. Supernatant samples were filtered using (Nanosep 3K Omega, Cytiva, Marlborough, MA, USA) microcentrifuge filter tubes followed by the addition of internal standards (Chenomx, Inc., Edmonton, AB, Canada) prior to their transfer to nuclear magnetic resonance (NMR) tubes for profiling. Proton spectra were obtained from the plasma aliquots of these samples using a Varian INOVA 600 MHz NMR (Chenomx, Inc., Edmonton, AB, Canada). The identification and quantification of metabolite peaks were carried out using the Chenomx NMR Suite 6.0 (Chenomx, Inc., Edmonton, AB, Canada).

### 2.3. Analysis

*T*-test comparisons between fibrosis and body mass index (BMI) were calculated with Welch’s *t*-test, given the unequal variance between the variables with a significance cutoff of *p* < 0.05. A systematic, unsupervised analysis of the individual datasets was performed with principal component analysis (PCA). This provided an unbiased assessment of the variation and the dependence (or independence) of the variables within each dataset. We proceeded to integrate the datasets in order to elucidate any potential associations between the different measurement modalities through the construction of MCNs. Briefly, significant pairwise Pearson correlations were used to construct the initial network maps of the variables (the nodes correspond to a measured variable and the links correspond to the statistically significant (*p* < 0.05) correlations). More explicitly, for a set of individual measurements, *i* in *α* and *j* in *β*, with $\left\{\tau ,p\right\}\in R^{1}$, the connectivity, $\chi \in N^{ixj}$, is given by
(1)$$\chi (\alpha_{i},\beta_{j})=\left\{\begin{array}{cc} 1 & \mathrm{if}\;c\left(\alpha_{i},\beta_{j}\right)\geq \tau_{c}\;\&\;p<\sigma_{c},\\ -1 & \mathrm{if}\;c\left(\alpha_{i},\beta_{j}\right)\leq -\tau_{c}\;\&\;p<\sigma_{c},\\ 0 & \mathrm{otherwise}. \end{array}\right.$$
in which α and β correspond to any two of the four datasets (clinical profile, histopathology, MRI, and metabolomic), respectively, for all unique combinations (irrespective of order), such that α≠β; c(·,·) is the correlation function (e.g., Pearson’s correlation function). Since a small number of measurements involved ordinal variables (e.g., histopathology NAS scores), Spearman coefficients were also calculated, which confirmed that the same set of variables met the significance criterion. Subsequent filtering was performed to further refine the network based on connectivity and correlations (cutoff $\tau_{c}=0.3$) and significance (cutoff $\sigma_{c}=0.05$) (Appendix A). The connectivity coefficients for the individual nodes within each MCN are calculated as the diagonal elements of $abs\left(\chi \right)\cdot abs\left(\chi^{T}\right)$.

Data analysis and processing were performed with Mathematica (v12, Wolfram Research, Inc., Champaign, IL, USA), Matlab (R1026b, The MathWorks, Natick, MA, USA), and Python scripts (v3.0).

## 3. Results

The mean age of the subjects was 14 ± 2 years with a BMI of 36 ± 6 kg/m^2^ (Figure 2; see Appendix A for other clinical variables). The violin plots for the male and female cohorts are qualitatively similar for these measurements. A BMI greater than 30 kg/m^2^ is considered obese and over 40 kg/m^2^ is considered severe/massive obesity. The majority of subjects (37) were obese (BMI 30–40 kg/m^2^), in addition to 16 subjects who were massively obese. There is a reported correlation between fibrosis and BMI in the pediatric NASH population [19]. We tested a binary association for BMI versus fibrosis; the results were statistically significant by Welch’s *t*-test in both directions. For a cohort comparison of massively obese versus not massively obese BMIs, there was an associated difference in histopathologic fibrosis scores with a *t*-statistic of 2.4 (*p* = 0.03). Conversely, for fibrosis scores greater than or equal to 2 versus less than 2 (on a scale of 0–5, with 0 being no fibrosis), there was an associated difference in the BMI with a *t*-statistic of 2.5 (*p* = 0.03).

In order to characterize the main sources of variation and the dimensionality of the individual datasets, an unsupervised analysis of each dataset was performed. Principal component analysis (PCA) of the individual datasets offers an evaluation of the variation and interdependence of the variables within each data type (Figure 3) and also helps to identify the key variables accounting for most of the variations within the datasets. Interestingly, many of the individual variables are nearly aligned with orthogonal axes, suggesting that they are largely independent of one another, and in turn may track different pathophysiological processes.

Additionally, one can appreciate the dependence versus independence of each of the measurements within the datasets when comparing two or more combinations of measurements (‘alternative, equivalent groups of variables’). For example in the histopathological dataset, fibrosis, ballooning, and lobular inflammation will essentially provide the same amount of information as steatosis, ballooning, and lobular inflammation or fibrosis, ballooning, and NAS score. Interestingly, the different histopathological measurements had quantitatively and qualitatively different histograms (see, for example, Appendix A), further supporting the independence of these measurements, although when combined (e.g., with the NAS score), they began to more closely approximate a normal distribution. For the MRI dataset, the most informative group of measurements consisted of the mean fat percentage, high T2 signal, and low T2 signal. The top six principal components are highlighted for the metabolomics dataset (Figure 3, panels D and E). Interestingly, the top principal components are described by only three metabolites (glucose, citrate, and lactate). These metabolites are all ‘key’ or characteristic metabolites in central metabolism involving glycolysis and the citric acid cycle, notably the ‘decision point’ for the metabolism of pyruvate (whether it will be oxidized to enter the citric acid cycle or reduced to lactate). The next three principal components involve glycerol and amino acids. Glycerol and beta-hydroxybutyrate are closely linked to fatty acid metabolism (particularly tri-, di-, and monoglyceride fatty acids).

Observing that the independent dimensionalities of the datasets are relatively low, the next step was to integrate these four datasets into a single, comprehensive network, through the construction of MCNs (see Section 2.3). Moreover, 45 variables with 62 nominal significant associations were identified, enabling the construction of a multiparametric clinical MRI–histopathologic NMR network map of pediatric NAFLD. Looking at the entire network (all four datasets together) reveals a complex web of associations/interactions (Figure 4). Across all of the variables (network nodes) with significant associations, leptin had the highest connectivity coefficient (cc = 11) followed by AST (cc = 7) and fibrosis (cc = 6). Interestingly the nodes corresponding to the different types of measurement modalities (clinical, metabolic, histopathology, and MRI) are distributed throughout the entire network. There are no focal clusters in each data type, reflecting the potential interdependence among the different measurement modalities.

The interconnectedness of the multiparametric association network (Figure 4) necessitated a means to filter some of the nodes and linkages of the network in order to identify any clusters or sub-networks. Given the high connectivity of leptin and other clinical blood measurements, such as transaminases, in order to decompose the global network, remove potentially obfuscating linkages, and delineate meaningful sub-networks, the clinical variables were removed; only the MRI, NMR (plasma metabolomics), and histopathological data were focused on. The result is a set of three disjoint networks (Figure 5) corresponding to fibrosis, lobular inflammation/steatosis, and ballooning. Each of these sub-networks centered around histopathological features have associated MRI measurements and plasma metabolites (with a total 19 links/associations).

## 4. Discussion

Pediatric NAFLD is a multi-factorial disease with complex metabolic and structural alterations; the fact that it develops decades earlier in pediatric populations, in comparison to adult populations with different histological patterns of progression, suggests that this disease must be studied independently in these populations [20]. The use of different measurement modalities to make the initial diagnosis (i.e., biopsy as the gold standard) and other less-invasive measurements to track disease status longitudinally add to the confusion and potential confounding effects when the relationships between different measurement modalities are not known. The MCN analysis and network maps presented herein provide a context to understand different histopathological-based characterizations of pediatric NAFLD with corresponding MRIs, blood proteins, and metabolomic measurements. To our knowledge, such a map that integrates these data has not been performed in pediatric NAFLD. From the initial MCN network (Figure 4), the dominance of leptin is clearly apparent; however, it also obscures the appreciation of potential sub-networks built around different aspects of the pathophysiological state of the disease. The PCA results also indicated that the dataset variations could be accounted for by a small number of the measurements; the blood/clinical measurement dataset was filtered out and the resulting MCN revealed three independent sub-networks, each with an associated set of metabolites, histopathologic features, and MR markers (Figure 5). These maps set the stage for characterizing different aspects of NAFLD pathophysiology and how they may relate to medical interventions, disease status/progression, and prognosis.

Different surrogates may be more helpful for tracking different aspects of disease; for example, fibrosis may potentially be tracked via high ADC (MRI) and isobutyrate, whereas steatosis may be more directly tracked through fat percentage (MRI) and/or transaminases (AST/ALT). Studies evaluating the efficacy of different therapeutic drug treatments for NAFLD have identified differences in treatment responses in terms of transaminase levels versus fibrosis versus steatosis versus NAS [21,22,23,24]. The long-term morbidity complications from NAFLD largely yield from sequelae related to late-stage fibrosis [3]; however, other histopathological components, such as steatosis and ballooning, are also important. The metabolic states/alterations over the course of the disease are not uniform derangements; they involve shifts in different metabolic states. For example, later cirrhotic stages involve inflammatory processes and reactive oxygen species (ROS) production, whereas earlier steatotic states involve adaptive mitochondrial mechanisms (antioxidant defense, mitophagy, and mitochondria biogenesis) [25].

In metabolic investigations on the pathogenesis and progression of NAFLD, lipid metabolism is commonly focused on [11,26]; there is evidence that metabolic alterations involve more than just lipid and ketone metabolism, but also glycolysis, the citric acid cycle, the urea cycle, and purine metabolism [14]. Ketone bodies, particularly β-hydroxybutyrate, have been associated with inflammasome inhibition and IL-18 production [14]; the MCN map implies an indirect negative correlation with fibrosis, which is consistent with non-alcoholic steatosis-related patterns of fibrosis [27]. The association between fibrosis, isobutyrate, and L-glutamate has also been previously reported [28]. Interestingly, acetate and hydroxybutyrate were not found in the same subnetworks (Figure 5), with the former being associated with ballooning. Urea was not directly measured in the metabolite panel; however, the negative associations between creatine and L-arginine and lobular inflammation indirectly implicate the urea cycle, which is noted to be disrupted in NAFLD/NASH [14].

In our cohort, we evaluated a pediatric population with histologic NAFLD diagnoses; thus, the focus was on identifying potential metabolite correlates with different stages of progression, as opposed to differentiating normal versus NAFLD individuals. Additionally, our study focused on the NMR profiling of core metabolite sugars, amino acids, and organic acids, and did not include fatty or bile acids. The associations between different amino acids with different histological features for the different subnetworks (Figure 5) is interesting, but potential mechanistic pathways are not immediately obvious (e.g., L-arginine was negatively correlated with lobular inflammation while fibrosis was negatively correlated with isobutyrate but positively correlated with L-aspartate). Thus, these findings provide an area to test in other independent datasets and further explore potential causal mechanisms. The interpretation challenge is contributed to by the fact that metabolomic plasma measurements reflect changes from the entire body (including the microbiome) as opposed to liver metabolomics. Measurements enabling targeted, tissue-specific metabolomics, such as clinical MR spectroscopy, may help address this in the future.

The NAS score is a summation of four individual measurements: steatosis, lobular inflammation, ballooning, and fibrosis. Different patterns of disease have been noted in the development and changes during the progression of the disease and have, in turn, led to different types and subtypes of the disease being recognized, including the differentiation between pediatric and adult manifestations of the disease. Apart from the epidemiological differences between adult and pediatric NASH, there are histopathological differences. Notably, pediatric NASH involves portal fibrosis, portal inflammation (more frequently than lobular inflammation), and periportal steatosis in contrast to adult NASH, which involves perisinusoidal fibrosis, lobular inflammation, and perivenular zone steatosis [29]. Ballooning is also apparently less commonly seen in the pediatric NAFLD population. These differences, notably steatosis with ballooning and perisinusoidal fibrosis, along with sparing of the portal tracts, contrast with steatosis accompanied by portal inflammation and fibrosis but without ballooning. The former has been referred to as adult (NASH type 1) and the latter as pediatric (NASH type 2) [29].

Since the constituent measurements comprising the NAS score do not move in tandem together, the different patterns of histopathologic alterations have contributed to the changing landscape of non-alcoholic liver steatosis diseases [30]. In addition to changing/revising the nomenclature due to new patterns of histopathological descriptions, there are also considerations regarding nomenclature and potential stigmatization results from the description of the disease. In 2020, an international expert consensus group proposed the name metabolic dysfunction-associated fatty liver disease (MAFLD) for non-alcoholic liver disease, as defined by hepatic steatosis, with at least one other factor, such as obesity, type 2 diabetes mellitus, or other metabolic dysregulation evidence [31]. An important point highlighted by the new name is that MAFLD is a standalone disease, and its diagnosis does not exclude the coexistence of other potential contributors to liver dysfunction [32]. More recently, a consensus study of over 200 hepatologists, gastroenterologists, pediatricians, endocrinologists, and hepatopathologists representing the American Association for the Study of Liver Diseases, the European Association for the Study of the Liver, and the Asociacion Latinoamericana para el Estudio del Higado, voted on revising the nomenclature to replace the NAFLD/MAFLD abbreviation with metabolic dysfunction-associated steatotic liver disease (MASLD), and in the process, maintained the relevant pathophysiological descriptor (e.g., steatosis) without any associated stigmatizing descriptors [33].

In our study, MCNs have shown that individual histopathologic features comprising the NAS score are correlated with different MR and metabolomic alterations. Treatments targeted at different phases of the disease may have different effects dependent on the degree/stage of progression in NAFLD; thus, it may be prudent to consider changes that may ameliorate the current stage, but also counteract progression to the subsequent stage. Given the significant role that metabolic dysfunction plays in pediatric NAFLD, diet and lifestyle changes will likely play important roles in treating the disease [29,32,34]. This multi-modality map of pediatric NAFLD provides a context for potentially understanding the different effects of medications on the disease course and progression, such as vitamin E versus PPAR-γ agonists, chemokine receptor antagonists, and other emerging medications [6,22,24]. In order to achieve these applications, further exploration of the potential causality between the links needs to be explored.

One potentially promising area of active research involves glucagon-like peptide-1 (GLP-1), which may be able to treat multiple aspects of NAFLD/MAFLD. An increasing number of studies have been exploring the application of glucagon-like peptide-1 receptor agonists (GLP-1 RAs) in patients with NASH/NAFLD, as type 2 diabetes and obesity are so highly associated with this disease [35]. While the outcome measures in these different trials have varied (from weight reduction to hepatic steatosis reduction to liver enzyme changes), the use of GLP-1 RAs to treat the spectra of non-alcoholic fatty liver diseases has shown some encouraging and interesting results. It provides a unique potential to treat multiple histopathological and clinical manifestations of nonalcoholic liver-associated dysfunction. A meta-analysis by Mantovani et al. [36] recently evaluated the current state of randomized clinical trials for GLP-1 RA for the treatment of NAFLD/NASH, focusing on six placebo-controlled and five active-controlled studies. Although the results were mixed for study outcomes based on liver enzymes, a preponderance of studies showed improvement in steatosis and histologic resolution of NASH [36].

A few additional areas in which further investigations are warranted include, (1) employing genome-scale metabolic networks for the mechanistic analysis of metabolism with omics data [37,38,39] and assessing correlations with the other measurements described (e.g., MRI and histopathological measurements), (2) incorporating other measurement modalities, such as PET/MR, as a means for non-invasive metabolomics [40], and (3) further exploring and developing other non-invasive measurement modalities, such as MR and ultrasound elastography [41]. Finally, it is important to recognize that disease progression does not follow a linear course and these maps are likely not static structures, but dynamic, with differences in the links potentially affected by particular genotypes as well as treatment modality and responses. For example, although the NAS score is a linear composition of its individual constituent measurements, each individual measurement varies along the course and progression of the disease in a non-linear fashion; thus, any analysis that focuses on identifying linear correlations over time will have difficulty in generalizing to all possible pediatric NAFLD populations. Thus, any “fixed” statistically-based association will likely change over time. Looking forward, future work exploring dynamic MCNs to identify robust associations versus transient ones will be of great interest. This approach could inform the pathologic course of NAFLD and provide further insight into strategies for diagnosis, tracking, and treating NAFLD/MAFLD/MASLD.

## 5. Conclusions

In summary, we presented a top-down analysis across multiple, disparate data types that was enabled through the construction of MCNs, which in turn provided insights into the complementary and independent nature of different types of measurements. The results have implications for the phenotypic characterization of MAFLD/MASLD, as well as provide strategies for the longitudinal evaluation of disease progression and treatment response.

## Figures and Tables

**Figure 1 metabolites-13-00929-f001:**
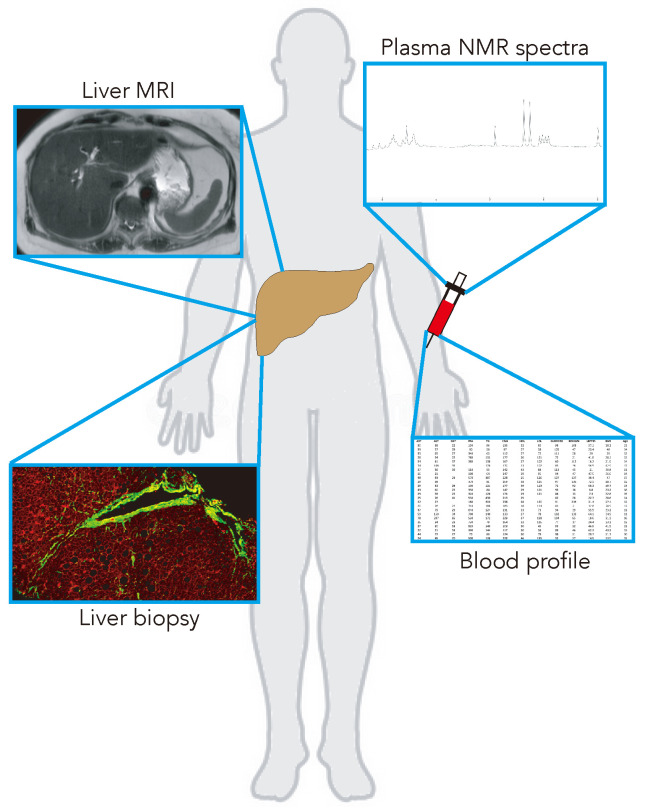
Four different modalities and measurements (ranging from non-invasive MR to blood draws and minimally invasive procedures) were obtained from pediatric subjects with NAFLD.

**Figure 2 metabolites-13-00929-f002:**
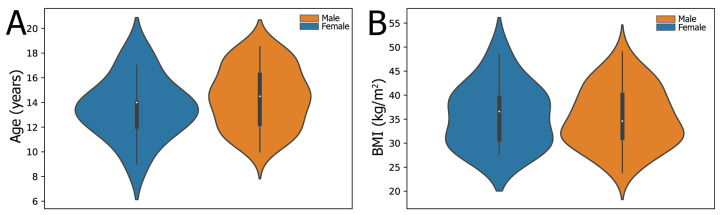
Violin plots for the age (**A**) and BMI (**B**) of the participants.

**Figure 3 metabolites-13-00929-f003:**
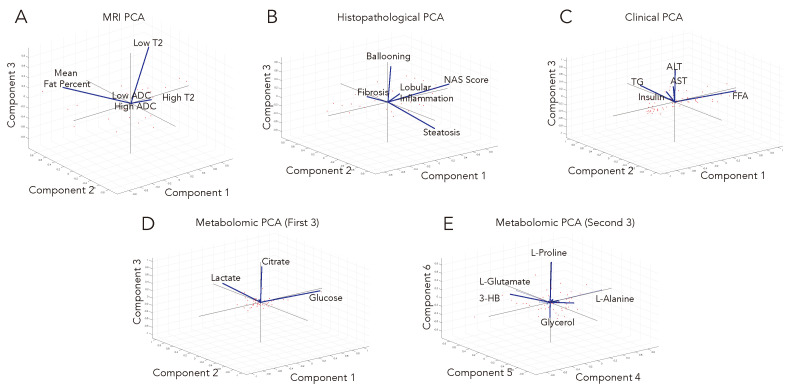
PCA was used to characterize the structure and variation among variables within each of the four datasets: (**A**) MRI, (**B**) histopathological variables, (**C**) clinical blood tests, and (**D**,**E**) metabolomics. The first three principal components account for at least 94% of the variation in each of the individual datasets. Principle components are commonly composed of linear combinations involving numerous variables; however, interestingly, these align well with just a few variables for each of the datasets. For example, in clinical data, the transaminases (ALT and AST), free fatty acids, and triglycerides are nearly mutually orthogonal. In a similar vein, the histopathological measurements (NAS score, fibrosis, and ballooning) are nearly orthogonal. Abbreviations: 3-HB: 3-hydroxybutyrate, ALT: L-alanine aminotransferase, AST: L-aspartate aminotransferase, FFA: free fatty acids, TG: triglycerides.

**Figure 4 metabolites-13-00929-f004:**
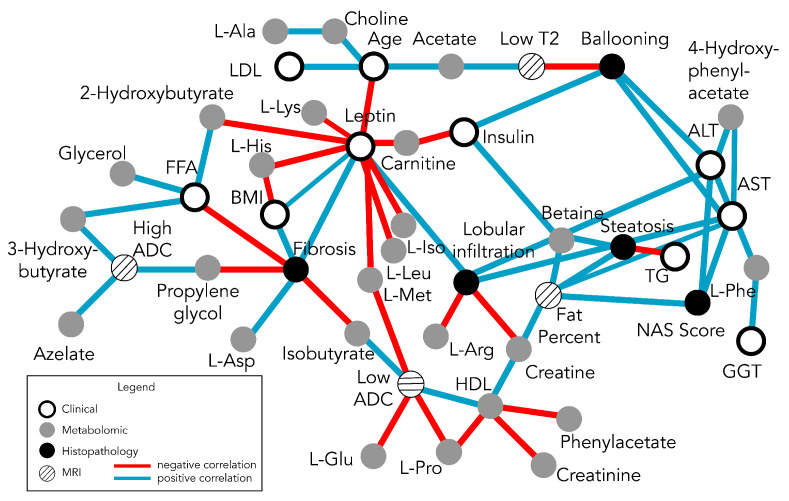
Fully integrated network map built upon nominal significant associations among clinical, MRI, histopathological, and metabolomic measurements. Each node in the network corresponds to measured variables and links between the nodes, reflecting significant correlations. The correlation directions (blue versus red for positive versus negative, respectively) and the number of different correlations for each node provide a glimpse into how the histopathological features relate to the plasma metabolomics, quantitative MRI features, and clinical blood measurements. Leptin provides the highest level of connectivity with a positive correlation with lobular inflammation but negative correlations with numerous metabolites, including amino acids, carnitine, and organic acids (hydroxybutyrates and acetate along with its associated derivatives). The histopathology features are largely independent, as highlighted by PCA (Figure 3), and are correlated with different groups of metabolites. The transaminases link steatosis, NAS, ballooning, and lobular inflammation. The remaining network nodes are connected to the rest of the network via fibrosis.

**Figure 5 metabolites-13-00929-f005:**
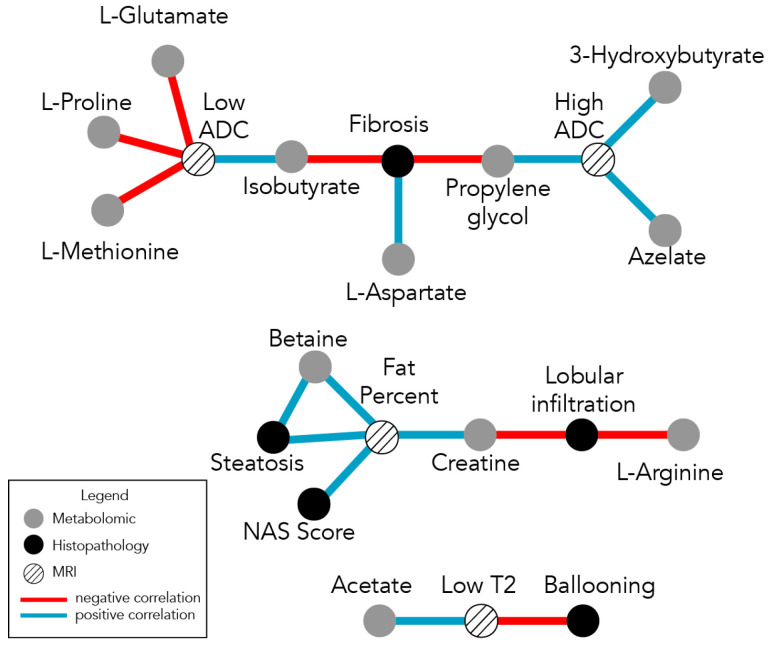
Network construction of the MRI, histopathology, and metabolomic datasets reveals three disjoint sub-networks centered around fibrosis, fat percentage, and ballooning, respectively. These three sub-networks provide a context to appreciate the underlying structure of the global network in Figure 4. The sub-networks support the concept that the progression of this disease involves different states and some variables will track some of the stages (e.g., primary inflammatory versus steatosis versus fibrosis) more closely than others; some variables will be more specific for particular stages of the course of disease progression. For example, ballooning and steatosis (observed to be nearly orthogonal to one another in the PCA plots in Figure 1B) are in different sub-networks. Analogously, the T2 signal and fat percentage on the MRI are independent in the PCA plots and are in different sub-networks. However ballooning and T2 are co-correlated, as are fibrosis and fat percentage. Vertices, link annotations, and labels are described in Figure 4.

## Data Availability

The data presented in this study are available in the article and upon request from the authors.

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
