# Peer review of "Multi-Modality, Multi-Dimensional Characterization of Pediatric Non-Alcoholic Fatty Liver Disease"

_metabolites, 2023, doi:10.3390/metabo13080929_

Round 1

Reviewer 1 Report

I congratulate the authors on a well written manuscript. The complex topic was clearly explained, its significance to a clinical audience was clear and the flow of the study was easy to follow. I do have some recommendations for improvement:

1. The study is based on data from paediatric NAFLD, the authors in several places state there are differences to adult disease. For non-hepatologists it would be helpful to state some of the key differences, for example histological changes. This would help put the data in to context.

2. The multi-modal modelling was very interesting and the authors state that it could identify nodes of interest at different stages of disease. Again for non-hepatologists can they state features that define the progression of the disease. This would help in interpreting the data, so in their analysis did they pick out features that were representative of specific disease stages? This was not obvious to this reviewer.

3. There are a few grammatical or typographical errors that need changing:

Line 31: non-invasive is repeated 

Line 43: change diagnosis to diagnose

Line 48: change exploring to explore

Line 129: change link to linked

Line 154: insert than before in adult populations

Line 190: remove has been

Line 193 change have to has

Reviewer 2 Report

Nonalcoholic fatty liver disease (NAFLD) is an important medical problem due to its high prevalence and inadequate diagnosis. NAFLD in children is a particular problem, which increases the interest of clinicians in its early diagnosis.

Comments:

1. The article practically lacks clinical characterisation of patients. It is recommended to add a table with clinical and demographic data, including the presence of comorbidities (especially endocrine diseases such as diabetes mellitus), it is recommended to add a table with data of laboratory tests, including glucose levels, etc.). 

2. It is recommended to add the date of Ethical Committee approval of the study protocol.

3. What medications were the patients taking?

4. It is recommended to improve and structure the abstract.
